# Evolutionary Diversity of Bat Rabies Virus in São Paulo State, Brazil

**DOI:** 10.3390/v17081063

**Published:** 2025-07-30

**Authors:** Luzia H. Queiroz, Angélica C. A. Campos, Marissol C. Lopes, Elenice M. S. Cunha, Avelino Albas, Cristiano de Carvalho, Wagner A. Pedro, Eduardo C. Silva, Monique S. Lot, Sandra V. Inácio, Danielle B. Araújo, Marielton P. Cunha, Edison L. Durigon, Luiz Gustavo B. Góes, Silvana R. Favoretto

**Affiliations:** 1Departamento de Produção e Saúde Animal, Faculdade de Medicina Veterinária de Araçatuba, UNESP—Universidade Estadual Paulista, Araçatuba 16050-680, Brazil; cristiano.carvalho@unesp.br (C.d.C.); w.pedro@unesp.br (W.A.P.); 2Institut Pasteur de São Paulo—IPSP, São Paulo 05508-020, Brazil; lgbgoes@usp.br; 3Programa de Pós-Graduação Interunidades em Biotecnologia, Laboratório de Virologia Clínica e Molecular, Departamento de Microbiologia, Instituto de Ciências Biomédicas, USP—Universidade de São Paulo, São Paulo 05508-000, Brazil; daniellebastos@yahoo.com.br (D.B.A.); srfavoretto@usp.br (S.R.F.); 4FAPESP Scholarship, Faculdade de Medicina Veterinária de Araçatuba, UNESP—Universidade Estadual Paulista, Araçatuba 16050-680, Brazil; marissolcl@hotmail.com (M.C.L.); sandra.valeria@unesp.br (S.V.I.); 5Secretaria de Agricultura e Abastecimento, Instituto Biológico de São Paulo, São Paulo 04016-035, Brazil; sequetin_cunha@hotmail.com; 6Agência Paulista de Tecnologia dos Agronegócios, Unidade de Pesquisa e Desenvolvimento de Presidente Prudente, Presidente Prudente 19015-970, Brazil; avealbas@yahoo.com.br; 7CNPq Technical Assistance Fellowship, Faculdade de Farmácia, UNIP—Universidade Paulista, Campus Araçatuba, Araçatuba 16018-555, Brazil; 8Núcleo de Pesquisas em Raiva—NPR, Departamento de Microbiologia, Instituto de Ciências Biomédicas, USP—Universidade de São Paulo, São Paulo 05508-000, Brazil; eldurigo@usp.br; 9Laboratório de Bioinformática e Virologia, Departamento de Genética, Evolução, Microbiologia e Imunologia, Instituto de Biologia, UNICAMP—Universidade de Campinas, Campinas 13083-862, Brazil; mdpcunha@unicamp.br; 10Instituto Pasteur—Secretaria da Saúde do Estado de São Paulo, São Paulo 01311-000, Brazil

**Keywords:** rabies virus, non-hematophagous bats, viral diversity, antigenic and genetic characterization

## Abstract

The history of the rabies virus dates back four millennia, with the virus being considered by many to be the first known transmitted between animals and humans. In Brazil, rabies virus variants associated with terrestrial wild animals, marmosets, and different bat species have been identified. In this study, bat samples from different regions of São Paulo State, in Southeast Brazil, were analyzed to identify their genetic variability and patterns. A total of 51 samples were collected over ten years (1999–2009) and submitted to the immunofluorescent technique using monoclonal antibodies for antigenic profile detection (the diagnostic routine used in Latin American countries) and genetic evolution analysis through maximum likelihood approaches. Three antigenic profiles were detected: one related to the rabies virus maintained by hematophagous bat populations (AgV3), part of the monoclonal antibody panel used, and two other profiles not included in the panel (called NC1 and NC2). These antigenic profiles were genetically distributed in five groups. Group I was related to hematophagous bats (AgV3), Groups II and III were related to insectivorous bats (NC1) and Groups IV and V were also related to insectivorous bats (NC2). The results presented herein show that genetic lineages previously restricted to the northwest region of São Paulo State are now found in other state regions, highlighting the need for a comprehensive genetic study of bat rabies covering geographic and temporal space, through expanded genomic analysis using a standard genomic fragment.

## 1. Introduction

Rabies is one of the most important viral infectious diseases and, with a history dating back four millennia, is considered by many to be the first known disease transmitted between animals and humans [1]. Rabies was initially described in humans and carnivores, but studies of bats and rabies in Brazil and Trinidad in the 1920s and 1930s showed the existence of a rabies virus aerial cycle. The existence of the aerial cycle explained epidemics that occur without the presence of carnivorous animals and how the virus continues to circulate in places where rabies in domestic animals has been controlled, showing the interrelationship of this aerial cycle with the terrestrial cycle [1,2,3].

The rabies virus (RABV) belongs to the Rhabdoviridae family and *Lyssavirus* genus, which contains 18 viral species [4], most of them associated with bats from the Old World. The *Lyssavirus rabies* species is the only one that circulates among numerous mammals, including bats, carnivores, and nonhuman primates, such as marmosets in Brazil [5,6,7]. Cross-species transmission has been observed among non-canid carnivores, bats, and other mammal species, leading to the emergence of new lineages also related to American bats [5].

Antigenic characterization studies have been conducted in several Latin American countries using a panel of monoclonal antibodies (MAbs) produced by the Centers for Disease Control and Prevention (CDC), Atlanta, USA. The use of these MAbs, from the 1980s onwards, established a new era in the knowledge of RABV reservoirs and transmissibility, resulting in immediate advances in epidemiological surveillance in those countries before laboratories implemented sequencing capability. Over the years, this tool has identified several other antigenic profiles not included in the original panel [6] established by Diaz et al. [8]. These additional profiles were detected in many countries, including Brazil [9,10,11,12,13,14,15]. In a certain way, this was already expected, considering that among the samples analyzed for establishing the panel profiles, there were no varieties of isolates from different South American bat species. This identification technique continues to be used routinely in most Latin American countries.

The genetic analysis of the N gene, with a chosen genome fragment (between position 1157 and 1476 of PV-NC_001542), not only allowed the correlation of host species with their geographic distributions but also confirmed the differences observed in the reactivity pattern in the antigenic tests among samples associated with different species of bats in different countries [9,16,17,18].

In Brazil, the first antigenic [11] and genetic [19] studies showed that, as in other Latin American countries, the two predominant antigenic variants/viral lineages were associated with viruses maintained by dogs (called AgV1 and AgV2) and the *Desmodus rotundus* variant (AgV3). Later, antigenic variants/viral lineages associated with terrestrial wild animals, such as foxes and wild dogs [20,21], marmosets [6], and different species of insectivorous and frugivorous bats, were identified [13,22,23,24,25,26].

Both the genetic and antigenic characterizations of bat RABV isolated in the State of São Paulo in southeastern Brazil have shown the existence of variants/lineages related to *D. rotundus* (AgV-3), *Tadarida brasiliensis* (AgV4), and *Lasiurus* sp. (AgV6), also circulating in other species of insectivorous and frugivorous bats, along with other antigenic profiles that were not included in the CDC monoclonal panel [6,8,10,21,22].

In this study, bat samples from different regions of São Paulo State were analyzed using evolutionary approaches to identify genetic variability of the RABV in São Paulo. These isolates are compared with isolates described in previous publications, characterized antigenically and genetically, for which sequences have been deposited in Genbank (Table A1, Appendix B).

## 2. Materials and Methods

### 2.1. Samples

This study included RABV isolates from 32 municipalities in different administrative regions of São Paulo State in Southeast Brazil (Figure 1), between 1999 and 2010.

All the samples had been previously diagnosed as positive for rabies by means of the fluorescent antibody test (FAT) [27] and mouse inoculation test (MIT) [28], considered the gold standard at the time of receiving samples for diagnosis between 1995 and 2010. In all, 48 bat samples were studied: 13 frugivorous (11 *Artibeus lituratus*, 01 *Artibeus planirostris*, 01 *Artibeus fimbriatus*) and 35 insectivorous (04 *Myotis nigricans*, 10 *Neoeptesicus furinalis*, 03 *Neoeptesicus diminutus*, 01 *Neoeptesicus* sp., 05 *Molossus fluminensis*, 03 *Molossus molossus*, 01 *Cynomops abrasus*, 02 *Nyctinomops laticaudatus*, 01 *Nyctinomops macrotis*, 01 *Lasiurus blossevillii*, 01 *Lasiurus ega,* 01 *Eumops glaucinus* and 02 non-hematophagous (NH) bats not identified). Samples from one cat, one bovine, and one horse were also included, giving 51 samples in total (Table 1).

### 2.2. Antigenic Characterization

Antigenic characterization was performed using the CDC (Atlanta, GA, USA) monoclonal antibodies (MAbs) panel according to the protocol determined by Favoretto et al. [11]. These eight MAbs against the RABV nucleoprotein, developed by CDC, can identify different RABV variants through different reactivity patterns.

### 2.3. Sequencing and Genetic Characterization

For genetic characterization, the initial steps (i.e., extracting RNA and obtaining cDNA) were performed using methods described in previous studies [13]. Molecular reactions were performed using primers described previously by Smith et al. [32] and Campos et al. [33] to amplify 320 base pairs from the coding and non-coding region of the nucleoprotein (between positions 1157 and 1476 of PV-NC_001542). Double-strand PCR-amplified products were purified using the ExoSAP-IT system (GE Healthcare Bio-Sciences Ltd.—USB Corporation, Cleveland, OH, USA) according to the manufacturer’s instructions, and Sanger sequencing was performed as previously described by Campos et al. [33]. The excess dideoxynucleotide terminators were removed with the Applied Biosystems Big Dye XTerminatorTM Purification Kit (Applied Biosystems, Foster City, CA, USA), following the manufacturer’s recommendations. Purified samples were subjected to electrophoresis in POP6 polymer using an ABI-PRISM model 3100 automatic sequencer (Applied Biosystems, Foster City, CA, USA). The samples were tracked automatically using the Automatic DNA Analyzer software package of the ABI-PRISM model 3100.

### 2.4. Phylogenetic Analysis

The obtained nucleotide sequences were pre-analyzed using the BLASTn program (https://blast.ncbi.nlm.nih.gov/Blast.cgi, accessed on 19 February 2025) to confirm amplification of the specific product and then aligned with available GenBank sequences (Appendix B) using Geneious Prime software version 2019.2.3. The chosen GenBank sequences were selected based on full information like the host, place, and year of collection. Pairwise distances were calculated by MEGA 11 version 11.0.13 (available at https://www.megasoftware.net/ accessed on 19 February 2025) and phylogenetic trees were reconstructed using IQTREE software version 2.4.0 (available at http://www.iqtree.org/ accessed on 18 February 2025). The best model fit determined by IQTree was TIM + F + I + G4. To analyze the temporal virus variability, we used TempEst v1.5.3 (available at http://tree.bio.ed.ac.uk/software/tempest/ accessed on 19 February 2025) in the phylogenetic tree and root-to-tip method with best-fitting root in correlation function. The time-scaled phylogenetic tree was analyzed via Augur version 21.1.0 and auspice version 2.62.0 implemented on Nextstrain [34,35].

### 2.5. Phylogenetic and Antigenic Site Amino Acid Visualization

Phylogenetic visualization of RABV sequences was conducted using R version 4.4.1 (14 June 2024, ucrt). A phylogenetic tree in nexus format was imported using the ape package and further processed and visualized with the ggtree, treeio, and igraph packages. The final annotated phylogenetic tree was visualized with tip points colored with regard to geographic location and a scale bar indicating time (years) and genetic distance (substitutions/site). The final figure editing and layout adjustments were performed using the Inkscape program version 1.3.2. The alignment used for phylogenetic tree reconstruction was translated to amino acid and used to prepare one Figure with the region of antigenic site I present in the nucleoprotein.

## 3. Results

Among the fifty-one samples submitted to antigenic characterization, twenty-five (49%) were characterized as AgV-3 (RABV maintained by *D. rotundus* hematophagous bat populations) and twenty-six were characterized as RABV maintained by non-hematophagous bat populations (NC1 and NC2); out of these, twenty-one (41.2%) presented the antigenic profile NC1 and five (9.8%) presented the antigenic profile NC2.

The samples from this study were segregated into five different phylogenetic groups, highlighted in colors according to their genetic and antigenic patterns (Figure 2). The group called Group I showed samples with a genetic lineage associated with the virus maintained by *D. rotundus* hematophagous bats and antigenic variant AgV3. The virus groups isolated from insectivorous bats presented four independent phylogenetic clades, called Group II, Group III, Group IV, and Group V, with the antigenic profile NC1 in Groups II and III, and antigenic profile NC2 in Groups IV and V. Group II presented the greatest diversity of its host species, consisting of *Eptesicus* spp. (currently called *Neoeptesicus*), *Eumops* spp., *Myotis* spp., *Nyctinomops* spp., and *Lasiurus* spp. Groups VI and VII, with isolates external to the present study, presented isolates from marmosets (Group VI) and bat isolates related to *Tadarida brasiliensis* (Group VII).

The lowest percentage of identity was observed in comparison with the clade related to marmosets (Group VI in the phylogenetic tree). Estimates of evolutionary divergence over sequence pairs between groups, obtained using the maximum likelihood method in MEGA 11, were used to calculate the distances between groups (Table A2, Appendix B), showing a range from 8.7% between Groups I and VII to 17.9% between Groups II and VI. Among the groups detected in this study, the highest within-group distances (Table A3, Appendix B) were observed in Groups II (6.7%), III (4.7%), and V (5.7%) while minor within-group distances were detected in Groups IV (1.8%) and I (2.5%).

In Figure 1, we can also observe the geographic distribution of the samples from this study and from Genbank, used for the reconstruction of the phylogenetic tree, according to the resulting groups (genetic characterization) and antigenic variants/profiles (AgV-3, NC1, and NC2).

During the analysis of the antigenic site present in the nucleoprotein, we identified genetic signatures for some groups in the phylogenetic tree. Groups I, IV, V, VI, and VII and the root group showed recognized patterns (AET, AEV, TEV, TEA, TEM, and TDV, respectively), indicating the stability of these genetic groups. On the other hand, we could not identify any pattern for Groups II (TEA, TDE, IDT, TEV, and TDV) and III (TEA, TEL, and TEV). In these groups, we found higher variability that was confirmed by tree topology and can be seen in Figure 3. The full map of the antigenic site was produced and can be accessed in the Appendix A.

The Pearson correlation coefficient calculated in the analysis of temporal virus variability in this dataset was 0.18 (*p* = 0.015), as shown in Figure 4. Although the correlation coefficient is slightly positive, the TempEst analysis showed the stable evolution rate to the dataset during the sampling period considered in the analysis.

## 4. Discussion

All 13 fruit bats of the genus *Artibeus* presented antigenic variant 3 (AgV-3), as did bats from the genera *Molossus* (04), *Eumops* (1), and *Neoeptesicus* (2) and non-identified bats (2). Similar results were observed in rabies-positive bat samples from other regions of São Paulo State [11,15,19,24,36]. Previous studies had demonstrated that the frequency of RABV in *Artibeus* was higher than that in *Desmodus* bats in the study area [36,37,38,39], which could explain how AgV-3 is present in species that do not co-inhabit with *D. rotundus* species. This is corroborated by the finding that the genetic lineage of *D. rotundus* is not exclusive to the species since this lineage has been detected in non-hematophagous bats such as the fruit bat *Artibeus lituratus* [40] and insectivorous bats in this study, in addition to other previous studies [13].

The samples antigenically described as NC1 in phylogenetic group II were segregated with samples from the same geographic region and with one sample (EU981922) from Uruguay, with a geographical distance of more than 1,200 miles. The samples in the phylogenetic group III were segregated with samples from the same geographical region and with one sample (AB297647) from Rio de Janeiro State, more than 300 m away, and another sample (AB618034) from Paraiba State, more than 1600 m away. This antigenic profile was previously described in São Paulo State [11,12,13,14,15,41]. These groups were previously related particularly with host species described by Oliveira et al. [42]; nonetheless, in the present study, we observed different species in the same clade/phylogenetic group. Bats play an important role in virus transmission and spread in the Americas [42,43,44], and it was clearly demonstrated in this study that insectivorous bats present a heterogeneous genetic distribution independent of host species.

The NC2 antigenic profile detected in five samples from the *Molossus* and *Lasiurus* genera was previously described in the same geographical region [7] and was observed in three monophyletic clades with high bootstrap value support (81 to 98%). In a previous study [13], two samples (GU646777 and HM854031) were segregated independently as subgroups; in the present study, these samples were segregated as part of phylogenetic groups IV and V, confirming the importance of including more sequences and information from other regions of the state and the country, as well as from other bat host species. This was also visualized on the map (Figure 1), where the geographical distribution of the variants was described. With the inclusion of new samples, a more homogeneous distribution of antigenic profiles and genetic lineages across Sao Paulo State can be observed since, previously, these profiles and lineages were restricted to the northwest region of the state [13]. Nonetheless, important data from previous publications regarding the central region of this state and from other states of Brazil could not be compared with the isolates from this study, considering that the authors sequenced a different genome region or only a coding region [15,40,42,45,46,47]. This reinforces the idea that the same region of the genome should be analyzed and standardized by researchers in future studies.

Currently, the term “antigenic variants” and the CDC MAbs panel still are used mainly in Latin America and only in a few accredited laboratories. The results obtained in the present sample’s dataset corroborated previous results [13,15] that showed that this panel, despite its valuable importance in the past, does not have sufficient resolution as high as that obtained using genetic tools to characterize RABV variants from non-hematophagous bats, leading to divergences. An approach of comparing samples from different geographical regions using antigenic and genetic characterization is no longer ideal, as previously highlighted in other studies [15]. In any case, the term ‘antigenic variant’ will not become extinct immediately since it is still the language used in Ministry of Health reports in Brazil, for example. As found in this study, the antigenic characterization was realized during sample processing almost twenty years ago, providing useful information that could be used for future studies and to better understand rabies epidemiology. Therefore, future RABV studies must be focused on genetic analyses to provide a deeper and more comprehensive understanding of the virus, explaining its epidemiology, its dynamics, and possible interventions. This could lead to significant advances in rabies surveillance, prevention, and control at both population and individual levels.

Antigenic site I was presented here using nucleoprotein amino acid alignments, and the authors observed a genetic signature that had previously been described [6]. However, for genetic groups II and III, these signatures could not be observed; in fact, they are genetically and antigenically diverse, independent of the comparison between these two approaches.

In the first RABV genetic studies in Latin America, it was established that the 320 nucleotides in the nucleoprotein carboxi-terminus region, including the nucleo- and phosphoprotein intergenic regions (non-coding) between genome positions 1157 and 1476 (based in PV genome NC_001542), could be the standard for phylogenetic studies because this region presents the large nucleotide variability, 1.9 times greater than in the coding region [17]. For phylogenetic analyses, according to Smith et al. [46], groups that present a distance higher than 5% from other groups can be considered a distinct genetic lineage. Thus, for this study, the groups in the phylogenetic tree were determined by following this consideration.

In the phylogenetic tree (Figure 3 and Appendix A), the high bootstrap values (100% for aerial cycle of transmission; 78% for genetic lineages segregated into Groups I, II, III and VII; and 81% for Groups I, II and VII) support the presence of basal genotypes of the virus. For example, sample HQ666860 from an insectivorous bat, *Neoeptesicus furinalis*, collected in 2009 presented a long branch in Group II (bootstrap 81%), indicating a high number of nucleotide substitutions. Group I, Group III, Group IV, and Group V showed bootstrap values of 100%, 98%, 78% and 92%, respectively; this tree topology could also explain how and why samples in the antigenic analysis from genetic groups II to V presented a different antigenic profile.

The positive but low correlation coefficient (0.18) associated with the phylogenetic analysis suggests that during the short period of sequence sampling, between the years of 1986 and 2022, there was an accumulation of diversity, but the occurrence of the common ancestor to all sequences was distant in the past. This had been previously shown by other researchers [42], and this result means a stable RABV evolution rate in the analyzed period. This result reflects the profile of zoonotic viruses such as rabies, considering only the aerial cycle, and agrees with the TMRCA (time to the most recent common ancestor) of approximately 170 years determined by de Souza et al. [40] when analyzing the *D. rotundus/A. lituratus* genetic lineage (both antigenically AgV3).

## 5. Conclusions

Despite some limitations, such as analyzing only a 320-nucleotide fragment; the limited number of sequences available in GenBank for this same fragment; the absence of relevant information such as on the date, species, and collection location of these available sequences in the GenBank; and the retrospective nature of this study performed over a decade ago, these data provide valuable insights into RABV among bats. The key findings of this study are as follows: (*i*) antigenic profiles and genetic lineages previously restricted to the northwest region of the state of São Paulo are now found in other state regions, (*ii*) future rabies studies must be focused only on genetic analysis, and (*iii*) there is a need for a comprehensive genetic study of bat rabies in São Paulo State and greater Brazil with diverse sample locations and expanded genomic analyses using a standard genomic fragment or full genome when possible. Moreover, focusing only on host species could lead to misleading conclusions about RABV evolution and dispersal concerning time and geography.

## Figures and Tables

**Figure 1 viruses-17-01063-f001:**
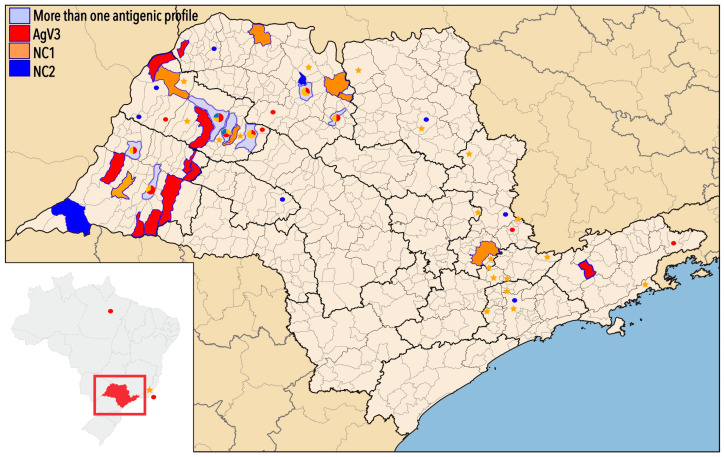
Geographic location of sample collection. Colors are related with antigenic variant/profile: red for Group I (AgV3) related to *D. rotundus*, orange for Group II and III (related to antigenic Non-compatible 1-NC1 profile), and dark blue for Groups IV and V (NC2). Cities where more than one genetic group and/or antigenic variant was detected are in pale blue with a graphic following determined pattern of colors. GenBank sequences used in phylogenetic tree reconstruction can be observed as red dots for Group I, orange stars for samples clustered in Groups II and III, and dark blue dots for Groups IV and V. The map was modified for this study using Inkscape software version 1.3.2 (available at www.inkscape.org accessed on 4 July 2025). The original map is available at https://pt.m.wikipedia.org/wiki/Ficheiro:SaoPaulo_MesoMicroMunicip.svg (accessed on 19 February 2025).

**Figure 2 viruses-17-01063-f002:**
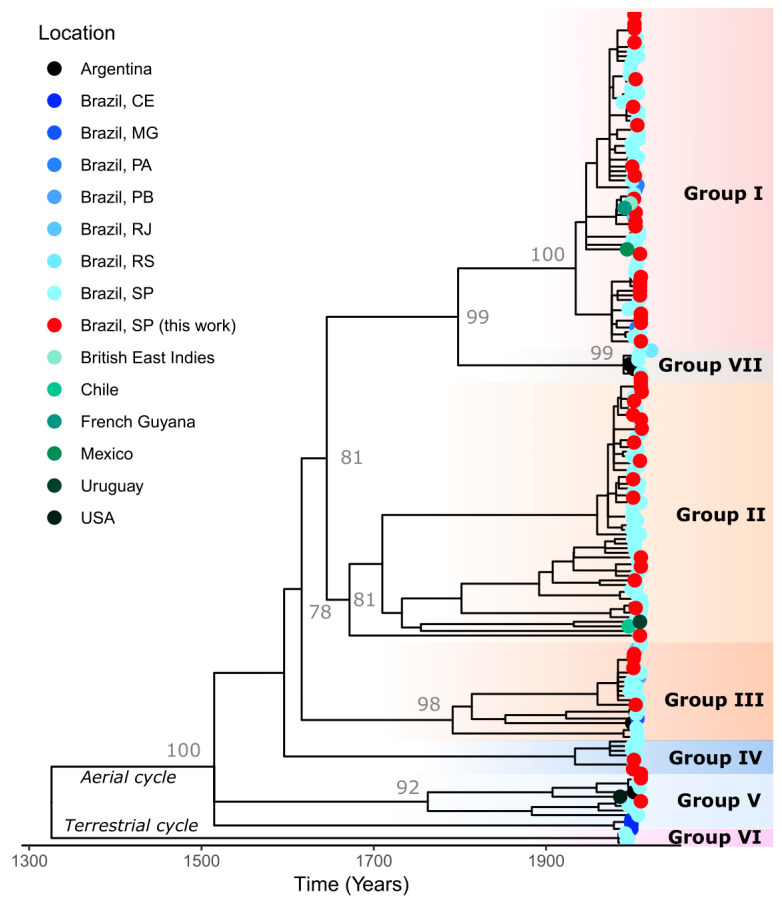
Time-scaled phylogenetic tree reconstructed using 320 nucleotides from nucleoprotein terminal gene using IQ-TREE software version 2.4.0, visualized and edited using FigTree software version 1.4.4 and plotted in RStudio version 2024.12.1+563 using ggtree version 3.14.0, treeio version 1.30.0, and igraph version 2.1.4 packages. The samples from this study can be observed in the tree in red dots; other samples from Brazil, available in GenBank, are marked in a blue pallet of colors while samples from other countries are marked in a green pallet of colors. The groups determined in this study are delineated vertically by red (Group I), orange (Groups II and III), and dark blue (Groups IV and V) shading. The groups without segregated samples from this study are shown with pink (Group VI) and gray (Group VII) shadings.

**Figure 3 viruses-17-01063-f003:**
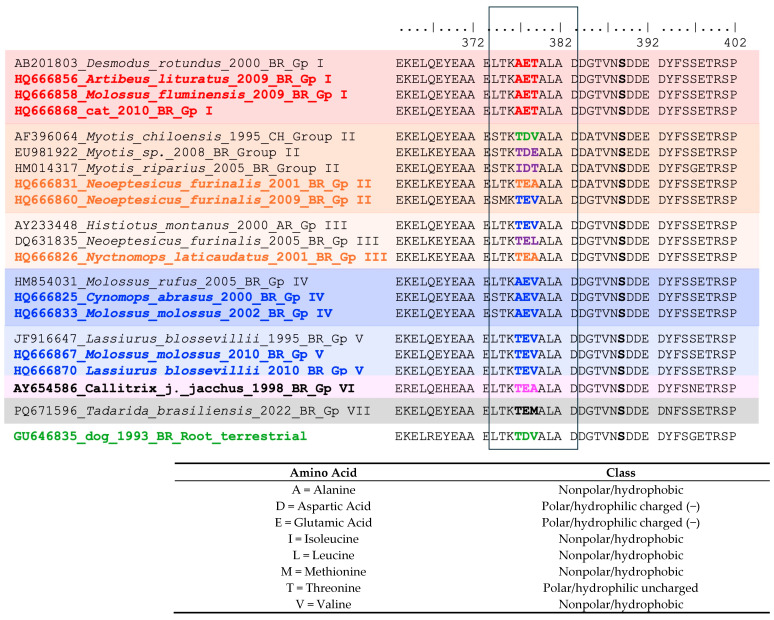
A partial amino-acid alignment showing the presence of antigenic site I in the nucleoprotein. The color match those in the phylogenetic tree presented in Figure 2. For this figure, the sequences used in the phylogenetic tree reconstruction were employed, maintaining only the variability in the antigenic site region, with a preference for sequences from this study. The genetic signature **AET** is shown in red (and also highlighted with red shading) for phylogenetic Group I related to hematophagous bat species *D. rotundus*; **AEV** is shown in dark blue (and also highlighted with dark blue shading) for phylogenetic Group IV related to non-hematophagous bats; **TEV** is shown in a gradient of dark blue (and also highlighted in a gradient dark blue shading) for phylogenetic Group V; **TEA** is shown in magenta/pink (and also highlighted with pink shading) for phylogenetic Group VI related to marmosets; **TEM** in black (and also highlighted with gray shading) for phylogenetic Group VII related to the bat species *T. brasiliensis*; **TDV** is shown in green for the root group related to the terrestrial cycle of transmission of RABV. For phylogenetic groups II and III (highlighted with gradient orange shading), it was not possible to find one genetic signature in the antigenic site I; in fact, in these groups, the variability was diverse and is noted in different colors: **TEA** in orange for the major antigenic site found (which was the same genetic signature found in Group VI related to marmosets), purple for variations (**TEL, TDE, IDT**), dark blue for the same signature present in Group V (**TEV**), and green for the same signature present in the terrestrial cycle of transmission (**TDV**) in one sequence available at GenBank (AF396064). The amino acid letters and class are outlined above.

**Figure 4 viruses-17-01063-f004:**
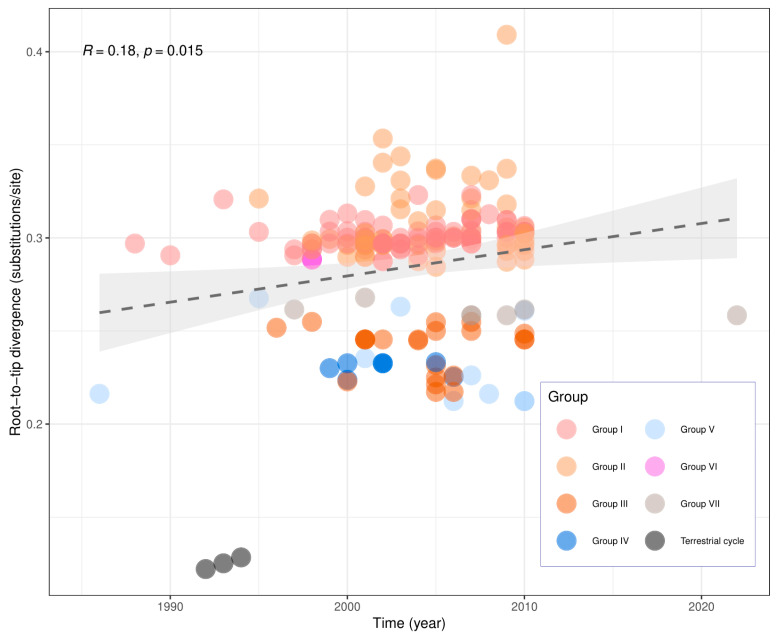
Correlation between genetic divergence and sampling time was obtained by a root-to-tip analysis using the RABV sequences and plotted using R command line. The dots are related with clades from phylogenetic tree, the red dots are related with Group I, the orange with Group II, the brown with Group III, the blue with Group IV, the clear blue with Group V, the pink with Group VI, the clear gray with the Group VII and the dark gray dots are related with the terrestrial cycle of transmission. The dashed line (colored dark gray) represents the regression line, and the light gray area around the dashed line is the confidence interval.

**Table 1 viruses-17-01063-t001:** Sequences obtained in the study, including environmental and geographical information.

GenBank Access	ID Sample/ Year	Species	Place of Origin	Antigenic Variant/Profile	Genetic Lineage
HQ666824	IB 346/99	*Artibeus lituratus*	São José do Rio Preto	V-3	*D. rotundus*
HQ666825	IB 777/00	*Cynomops abrasus*	Ipiguá	NC2	Insect. Bats
HQ666826	IB 249/01	*Nyctinomops macrotis*	São José do Rio Preto	NC1	Insect. bats
HQ666827	IB 250/01	*Nyctinomops laticaudatus*	São José do Rio Preto	NC1	Insect. Bats
HQ666828	IB 636/01	*Neoeptesicus furinalis* *	Olímpia	NC1	Insect. Bats
HQ666829	IB 808/01	*Artibeus lituratus*	São José do Rio Preto	V-3	*D. rotundus*
HQ666830	IB 1019/01	*Neoeptesicus* sp. *	Cardoso	NC1	Insect. Bats
HQ666831	IB 1070/01	*Neoeptesicus furinalis* *	São José do Rio Preto	NC1	Insect. Bats
HQ666832	IB 62/02	*Neoeptesicus furinalis* *	Catanduva	NC1	Insect. Bats
HQ666833	IB 109/02	*Molossus molossus*	Ilha Solteira	NC2	Insect. Bats
HQ666834	IB 835/02	*Myotis nigricans*	Cajobi	NC1	Insect. Bats
HQ666835	IB 992/02	*Artibeus lituratus*	Dracena	V-3	*D. rotundus*
HQ666836	IB 1021/02	*Molossus fluminensis* *	Presidente Venceslau	V-3	*D. rotundus*
HQ666837	IB 1141/02	*Neoeptesicus furinalis* *	Santo Anastácio	NC1	Insect. Bats
HQ666838	IB 1256/02	NH bat (not identified)	Martinópolis	V-3	*D. rotundus*
HQ666839	IB 1371B/02	*Artibeus lituratus*	Presidente Venceslau	V-3	*D. rotundus*
HQ666840	IB 1535/02	*Artibeus lituratus*	Taciba	V-3	*D. rotundus*
HQ666841	IB 1539/02	*Molossus molossus*	Presidente Prudente	V-3	*D. rotundus*
HQ666842	IB 1782/02	*Lasiurus ega*	Presidente Prudente	NC1	*D. rotundus*
HQ666843	IB 349/03	*Artibeus planirostris*	Santa Fé do Sul	V-3	*D. rotundus*
HQ666844	IB 350/03	NH bat (not identified)	Catanduva	V-3	*D. rotundus*
HQ666845	IB 791/03	*Artibeus lituratus*	Presidente Prudente	V-3	*D. rotundus*
-	IB 826/03	*Artibeus fimbriatus*	São José do Rio Preto	V-3	ND
HQ666846	IB 168/04	*Myotis nigricans*	Campinas	NC1	Insect. Bats
HQ666847	IB 184/04	*Nyctinomops laticaudatus*	São José do Rio Preto	NC1	Insect. Bats
HQ666848	IB 550/04	*Artibeus lituratus*	Caçapava	V-3	*D.rotundus*
HQ666849	LRU 329/05	*Eumops glaucinus*	Araçatuba	V-3	*D.rotundus*
-	LRU 397/05	*Artibeus lituratus*	Araçatuba	V-3	ND
HQ666850	LRU 43/09	*Neoeptesicus diminutus* *	Pereira Barreto	NC1	Insect. Bats
HQ666851	LRU 84/09	*Neoeptesicus diminutus* *	Araçatuba	NC1	Insect. Bats
HQ666852	LRU 149/09	*Myotis nigricans*	Coroados	NC1	Insect. Bats
HQ666853	LRU 181/09	*Artibeus lituratus*	Penápolis	V-3	*D. rotundus*
HQ666856	LRU 325/09	*Artibeus lituratus*	Birigui	V-3	*D. rotundus*
HQ666857	LRU 374/09	*Artibeus lituratus*	Guararapes	V-3	*D. rotundus*
HQ666858	LRU 389/09	*Molossus fluminensis* *	Guararapes	V-3	*D. rotundus*
HQ666859	LRU 433/09	*Myotis nigricans*	Penápolis	NC1	Insect. Bats
HQ666860	LRU 589/09	*Neoeptesicus furinalis* *	Penápolis	NC1	Insect. Bats
HQ666854	LRPP 199/09	*Neoeptesicus furinalis* *	Dracena	NC1	Insect. Bats
HQ666855	LRPP 224/09	*Neoeptesicus furinalis* *	Parapuã	V-3	*D. rotundus*
HQ666861	LRPP 672/09	Bovine/Cattle	Narandiba	V-3	*D. rotundus*
HQ666862	LRU 17/10	*Molossus fluminensis* *	Penápolis	NC1	Insect. Bats
KU299782	LRPP 28/10	Horse	Taciba	V-3	*D. rotundus*
HQ666864	LRPP 43/10	*Neoeptesicus diminutus* *	Osvaldo Cruz	V-3	*D. rotundus*
HQ666865	LRU 60/10	*Neoeptesicus furinalis* *	Birigui	NC1	Insect. Bats
HQ666866	LRU 76/10	*Neoeptesicus furinalis* *	Penápolis	NC1	Insect. Bats
HQ666867	LRU 169/10	*Molossus molossus*	Araçatuba	NC2	Insect. Bats
HQ666868	LRU 171/10	Cat	Araçatuba	V-3	*D. rotundus*
HQ666869	LRU 177/10	*Molossus fluminensis* *	Birigui	NC2	Insect. Bats
HQ666870	LRPP 198/10	*Lasiurus blossevillii*	Teodoro Sampaio	NC2	Insect. Bats
HQ666871	LRU 299/10	*Neoeptesicus furinalis* *	Penápolis	NC1	Insect. Bats
HQ666872	LRU 300/10	*Molossus fluminensis* *	Penápolis	V-3	*D. rotundus*

ND = not done; NC = not compatible; NH = non-hematophagous; V-3 = variant 3. IB—Rabies Laboratory of “Instituto Biológico de São Paulo”; LRU—Rabies Laboratory of UNESP (São Paulo State University), Araçatuba; LRPP—Rabies Laboratory of APTA (São Paulo Agribusiness Technology Agency) of Presidente Prudente; * new taxonomy bat species classification according to [29,30,31].

## Data Availability

The original contributions presented in this study are included in the article. Further inquiries can be directed to the corresponding author(s).

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
