# Peer review of "Evolutionary Diversity of Bat Rabies Virus in São Paulo State, Brazil"

_viruses, 2025, doi:10.3390/v17081063_

Round 1
Reviewer 1 Report
Comments and Suggestions for Authors
In this study, Queiroz et al. present results of partial sequencing of 48 isolates of bat-associated rabies virus (RABV) from the Brazilian state of São Paulo, collected between 1999 and 2009. To this end, 320 nucleotides of the nucleoprotein (N) carboxi-terminus region including the N and phosphoprotein (P) intergenic region (non-coding) between genome positions 1157 to 1476 were sequenced. Phylonegetic analysis revealed that the RABV isolates, which had three antigenic variants (AgV), i.e. AgV3, non-compatible AgV NC1 and NC2, represented five clades, some of which were widely separated and evolutionare stable.
The methodology used is technically sound and state of the art, but the rationale and objectives for conducting the study and the chosen approach of comparing antigenic and genetic traits is not very convincing.
Here some of my major concerns:
- Unfortunately, the study is an example of the adherence to traditional approaches to characterize terrestrial and especially bat-associated RABVs in Latin America to explain unclassifiable antigenetic reaction patterns with established Mabs. Based on current knowledge, generally, the comparison of antigenic and genetic traits is mainly of academic interest, but has no implications for bat rabies control and prevention in humans. It is well known that antigenic variants of RABV can consist of different finely branched phylogenetic clades whose genetic variations have no bigger influence on the coding proteins, which is why they exactly show the same or similar reaction patterns with MAbs during antigen characterization. If today sequencing is the generally recognized standard for characterizing RABVs to show their diversity, the question arises whether antigen characterization is still necessary at all and if so, why? This question is not being asked or even touched upon.
- Even if this comparative approach is used one should asked what new insights and information can be deduced from it? But then it must be chosen in such a way that it makes sense. In other words, it would make sense to sequence regions of the N protein encoding the antigenic sites (?) the CDC Mabs are binding to. However, the authors chose to sequence the N carboxi-terminus region including the N and P non-coding, intergenic region, which makes it very difficult to draw meaningful conclusions. Therfore it is not surprising that the observed genetic traits do not match with the obtained genetic clades and therefore, two clades my represent one antigenic variant as shown for NC1 and NC2 comprising clades II and III and IV and V, respectively. It is the authors themselves that come to the conclusion that a standardized approach, better to say a meaninful target region of the N gene, should be used in future studies.
- How many NCs are there? The two NCs referred to in the paper appear to have a very different reaction pattern with the available MAbs, with RABVs belonging to NC1 and NC2 reacting with MAbs C4, C10, C12 and C4, C9, C10 respectively. Why are they not referred to as new antigen variants V-4 or V-5, for example, and retain this confusing term “non-compatible”?
- Why is the MIT still used to confirm FAT positive samples, if for animal welfare reasons it is no longer state of the art and should be replaced by RTCIT?
- Please provide the genebank accession numbers of the RABV isolates from other countries, e.g. Chile, French Guiana, Mexico, Uruguay and the USA, that were included in this study. These can be included in the table. 1)
- 1: The color code should be provided in the map rather than in the figure capture. What is meant by pale blue?
- 2: The gradation of the color code on the left side is barely visible in the phylogenetic tree on the right side and sometimes does not stand out from the background coloration of the different clades.
- Please, use established technical terminology. Here some examples: (1) different wording, e.g. reactivity profile, antigenic profile, antigenic pattern, reaction pattern, antigenic variant, genotypes and lineages, are used for one and the same thing. (2) Whether you call the divisions created “clades” or “groups” is irrelevant, but for reasons of consistency, please, use one word throughout the text. (3) From an epidemiological point of view the term “airborn cycle” is unfortunate. Bats may be able to fly, but they, too, are terrestrial animal species. Either use “bat-mediated” or “ bat-associated”.
All in all, the authors must decide whether to focus only on the genetic characterization and phylogenetic analysis, omitting the comparison with the antigenic response patterns, or if they do not want to do this, more needs to be done to work out the dicrepancies in this case. In both cases, the introduction (no overview of global lyssavirus species so far) and discussion need to be improved; both are currently very bumpy.
Comments on the Quality of English Language- The english needs improvements. It would be good to seek help of a native speaker.
Author Response
Response to reviewer 1 major concerns:
1. Unfortunately, the study is an example of the adherence to traditional approaches to characterize terrestrial and especially bat-associated RABVs in Latin America to explain unclassifiable antigenetic reaction patterns with established Mabs. Based on current knowledge, generally, the comparison of antigenic and genetic traits is mainly of academic interest, but has no implications for bat rabies control and prevention in humans. It is well known that antigenic variants of RABV can consist of different finely branched phylogenetic clades whose genetic variations have no bigger influence on the coding proteins, which is why they exactly show the same or similar reaction patterns with MAbs during antigen characterization. If today sequencing is the generally recognized standard for characterizing RABVs to show their diversity, the question arises whether antigen characterization is still necessary at all and if so, why? This question is not being asked or even touched upon.
R. At the time our study was conducted (2008-2010) the Brazilian Official Rabies Laboratories (accredited by the Ministry of Health) were interested in determining the geographic location of these different antigenic variants/profiles and their epidemiological importance and the antigenic characterization by CDC MAbs was widely used. Even today, the technique continues to be used by reference laboratories in Brazil and other Latin countries. This issue was addressed in the text considering why this technique is still used in Latin-Americans countries (3rd paragraph of the Introduction and 4th paragraph of the Discussion).
2. Even if this comparative approach is used one should asked what new insights and information can be educed from it? But then it must be chosen in such a way that it makes sense. In other words, it would make sense to sequence regions of the N protein encoding the antigenic sites (?) the CDC Mabs are binding to. However, the authors chose to sequence the N carboxiterminus region including the N and P non-coding, intergenic region, which makes it very difficult to draw meaningful conclusions. Therfore it is not surprising that the observed genetic traits do not match with the obtained genetic clades and therefore, two clades my represent one antigenic variant as shown for NC1 and NC2 comprising clades II and III and IV and V, respectively. It is the authors themselves that come to the conclusion that a standardized approach, better to say a meaninful target region of the N gene, should be used in future studies.
R. We sequenced the N gene, with the chosen genome fragment (between position 1157 and 1476 of PV – NC_001542) because this region presents the large nucleotide variability with 1.9 times greater than in coding region*. The MAbs are directed to one of the three antigenic sites of rabies virus, that is located in this fragment. In the 5th paragraph of the Discussion, we mentioned why to use this specific fragment of the N gene.
- Kissi, B.; Tordo, N.; Bourhy, H. Genetic Polymorphism in the Rabies Virus Nucleoprotein Gene. Virology 1995, 209, 526–537, doi:10.1006/viro.1995.1285.
3. How many NCs are there? The two NCs referred to in the paper appear to have a very different reaction pattern with the available MAbs, with RABVs belonging to NC1 and NC2 reacting with MAbs C4, C10, C12 and C4, C9,
C10 respectively. Why are they not referred to as new antigen variants V-4 or V-5, for example, and retain this confusing term “non-compatible”?
R. Since the first antigenic characterization study was conducted in Brazil in the early 2000’s (Favoretto et al., 2002), patterns of reactivity to monoclonal antibodies that were not defined by the CDC panel have been identified, including the pattern of reaction to MAbs C4, C10, C12 and C4, C9, C10 identified in our study. These different patterns were called "non-compatible" and other authors from Brazil (cited in the manuscript) continue to adopt this nomenclature. These patterns occur in bat samples and are not exclusive to a given species, like other predefined variants, so they are not considered new variants. For the Latin American the panel includes 11 defined antigenic variants, according to the reactivity pattern (Mattos & Mattos, 1998) and cited by Menozzi et al., 2017.
Favoretto, S.R.; Carrieri, M.L.; Cunha, E.M.S.; Aguiar, E.A.C.; Silva, L.H.Q.; Sodre, M.M.; Souza, M.C.A.M.; Kotait, I. Antigenic Typing of Brazilian Rabies Virus Samples Isolated from Animals and Humans, 1989-2000. Rev. Inst. Med. Trop. Sao Paulo 2002, 44, 91–95, doi:10.1590/s0036-46652002000200007.
Mattos, C; Mattos C (1998) Uso de anticuerpos monoclonales para la tipificacio´n antigenica de aislamientos de virus rábico. In: OPAS OP de LS (ed) Consorc. la OPS Lab. Ref. en rabia las Américas, OPS. Whashington, DC, pp 2–11
MENOZZI, B. D. et al. Antigenic and genotypic characterization of rabies virus isolated from bats (Mammalia: Chiroptera) from municipalities in São Paulo State, Southeastern Brazil. Archives of Virology, v.162, n.5, p.1201-09, Jan. 2017. DOI: 10.1007/s00705-017-3220-9.
4. Why is the MIT still used to confirm FAT positive samples, if for animal welfare reasons it is no longer state of the art and should be replaced by RTCIT?
R. The MIT analysis was performed when that sample ware sent to RABV diagnosis between 1995 and 2010. At that time, the MIT was still used as a diagnostic routine (gold standard), today it is no longer used and has been replaced by virus isolation in cell culture.
5. Please provide the genebank accession numbers of the RABV isolates from other countries, e.g. Chile, French Guiana, Mexico, Uruguay and the USA, that were included in this study. These can be included in the table. 1)
R. This information about the genebank accession numbers of isolates from other countries are presented in Appendix A - TableA1 – bat samples sequences used in this work
6. 1: The color code should be provided in the map rather than in the figure capture. What is meant by pale blue?
R. We accepted his suggestion, and the color code are included in Figure 1 (map). The pale blue means places where more than one antigenic variant/profile were detected. This information is also in the caption.
2: The gradation of the color code on the left side is barely visible in the phylogenetic tree on the right side and sometimes does not stand out from the background coloration of the different clades.
R. Following the suggestion of reviewer 2, we updated the Figure 2 to a new time-scale phylogenetic tree with a change in the colors.
7. Please, use established technical terminology. Here some examples: (1) different wording, e.g. reactivity profile, antigenic profile, antigenic pattern, reaction pattern, antigenic variant, genotypes and lineages, are used for one and the same thing. (2) Whether you call the divisions created “clades” or “groups” is irrelevant, but for reasons of consistency, please, use one word throughout the text. (3) From an epidemiological point of view the term “airborn cycle” is unfortunate. Bats may be able to fly, but they, too, are terrestrial animal species. Either use “bat-mediated” or “bat-associated”. All in all, the authors must decide whether to focus only on the genetic characterization and phylogenetic analysis, omitting the comparison with the antigenic response patterns, or if they do not want to do this, more needs to be done to work out the dicrepancies in this case. In both cases, the introduction (no overview of global lyssavirus species so far) and discussion need to be improved; both are currently very bumpy.
R. We prepared a complete revision of the manuscript with the reviewer’s recommendation, including the MDPI English Editing Service, and most of these terms were changed.
Reviewer 2 Report
Comments and Suggestions for Authors
Rabies is a zoonotic disease of major public and animal health concern. Therefore, molecular epidemiological studies play a critical role in informing health services and guiding appropriate control measures.
The manuscript is well written, and the study design is sound. While the use of monoclonal antibodies (MAbs) may appear somewhat outdated, the authors appropriately validated their findings by comparing MAb results with sequencing and phylogenetic analysis. This approach is acceptable and adds value to the manuscript. The study provides important insights into the antigenic and genetic profiles of rabies virus strains circulating in Brazil, which will be of interest to the readership and makes a significant contribution to the field.
As a complementary suggestion, including amino acid sequence comparisons—particularly for key antigenic sites—alongside the MAb results would enhance the discussion. Additionally, constructing a time-scaled phylogenetic tree to illustrate viral evolution over time would further strengthen the manuscript.
Author Response
Response to reviewer 2
1. As a complementary suggestion, including amino acid sequence comparisons—particularly for key antigenic sites—alongside the MAb results would enhance the discussion. Additionally, constructing a time-scaled phylogenetic tree to illustrate viral evolution over time would furtherstrengthen the manuscript.
R. We accepted his suggestion and included a new figure (Figure 3) with the amino acid sequence comparisons related to the only viral antigenic site present in the sequenced fragment. We also included one extra figure (Figure S2) in supplemental material with the antigenic site for all sequences used in the phylogenetic tree reconstruction.
2. Concerning to the suggestion about the construction of a time-scaled phylogenetic tree
R. We really appreciate this suggestion and accepted. The updated new time-scale phylogenetic tree is in the main manuscript as Figure 2.
Round 2
Reviewer 1 Report
Comments and Suggestions for Authors
I'm really pleased with the revised version of the manuscript — the improvements in both the technical presentation and the clarity of the discussion and language are clear and significant. I truly appreciate the effort that went into this.
However, it is a little disappointing that the authors did not address my initial question regarding the parallel, comparative testing approach. Rather than merely explaining that this was a request by the ministry, I would have expected them to engage with the issue and, drawing on their expertise, offer a perspective in the Conclusions on whether antigen characterization of RABVs using monoclonal antibodies remains routinely and scientifically necessary or could be replaced by more up-to-date whole genome sequencing. I would appreciate the author could still do so.
There are still a few spelling mistakes.
Author Response
Resposta aos comentários do revisor X
|
1. Resumo |
|
|
|
Muito obrigado por dedicar seu tempo à revisão deste manuscrito. Abaixo, você encontrará as respostas previstas e as revisões/correções correspondentes destacadas/em acompanhamento de alterações nos arquivos reenviados .
|
||
|
2. Questões para Avaliação Geral |
Avaliação do revisor |
Resposta e Rejeição |
|
A introdução fornece contexto suficiente e inclui todas as referências relevantes? |
Sim |
Não aplicável |
|
Todas as referências citadas são relevantes para a pesquisa? |
Sim |
|
|
O desenho da pesquisa é protegido? |
Sim |
|
|
Os métodos estão descritos especificamente? |
Sim |
|
|
Os resultados são apresentados de forma clara? |
Sim |
|
|
As conexões são complementadas pelos resultados?
|
Sim |
|
|
3. Resposta ponto a ponto aos comentários e sugestões dos autores |
||
|
Comentários: 1) Na Introdução e em todo o manuscrito, por favor, exclua a referência a um ciclo "aéreo", pois dá a impressão de que os lyssavírus são encontrados no ar, assim como um ciclo "terrestre". Esta é uma terminologia confusa e antiga, especialmente para novos leitores. Embora os morcegos sejam voadores, eles são mamíferos terrestres! (ln 57 ff). Resposta 1 : Quando o revisor afirma que os morcegos são mamíferos terrestres, ele está correto, se considerarmos a categoria de mamíferos (terrestres e aquáticos). No entanto, quando usamos os termos aéreo e terrestre, estamos nos referindo ao ciclo de transmissão da raiva por mamíferos terrestres (canívoros, por exemplo) que se movem no solo e morcegos que se movem pelo ar (mamíferos voadores). Por esse motivo, não concordamos com a solicitação do revisor para excluir a referência a um ciclo 'aéreo'. O conceito de ciclo aéreo e terrestre está bem estabelecido e tem sido usado com frequência e em artigos atuais. Por exemplo, na Edição Especial Advances in Rabies Research 2024 da Viruses, foi publicado um artigo aceito em 30 de maio de 2025, usando essa terminologia ( https://www.mdpi.com/1999-4915/17/6/788 ). Também incluímos essa nova referência em nossa bibliografia. Outras referências que utilizam esta terminologia podem ser acessadas nos links a seguir: DOI: 10.3390/tropicalmed6020098 – Meske et al. 2021 DOI: 10.20506/rst.37.2.2808 – Gilbert, 2018 DOI: 10.1590/1678-4685-GMB-2019-0370 – Oliveira et al., 2020
2) Por favor, exclua a palavra "O" vírus da raiva, pois existe apenas um, taxonomicamente falando (Lyssavirus rabies), mas muitos, do ponto de vista biológico. Pode começar simplesmente com "O vírus da raiva pertence...". (Ln 61). Resposta 2 : Aceitamos o comentário do revisor. Excluímos a palavra solicitada.
3) Ao falar sobre Mabs em um sentido histórico, podemos relatar que este era um método rápido e tecnicamente relevante para países de baixa e média renda que não tinham capacidade de sequenciamento (Ln 72 ff). Resposta 3 : Incluímos essas informações no manuscrito.
4) Explique por que você ainda usa o MIT (Ln 115). Resposta 4 : Já respondemos em nossa última revisão e agora incluímos na linha 114-115 do manuscrito principal o motivo pelo qual essas técnicas foram utilizadas. Até 2024, os laboratórios brasileiros de diagnóstico da raiva ainda utilizavam o IM. Um artigo de pesquisadores brasileiros de 2023 também descreve o uso do MIT. DOI: https://doi.org/10.1002/jmv.29046 - de Souza et al., 2023
5) Em todo o MS, ao usar % (por exemplo, 41,2%), mude para decimais (ou seja, 41,2%) para todos os valores relevantes (Ln 181 ff). Resposta 5 : Atualizamos as vírgulas por ponto para corrigir a representação em %. Obrigado por apontar isso.
6) Em todo o MS, altere a expressão “raiva positiva” para “raivosa” (Ln 278). Resposta 6 : Pedimos desculpas pela falta de clareza. Reescrevemos a expressão e a alteramos para “mostras de morcegos com resultado positivo para raiva… (linha 272)” porque se refere às amostras de morcegos, não ao animal (morcego).
7) Você está falando sobre a frequência do RABV, não da doença em si (ln 279). Resposta 7 : Desculpas pelo erro. Reescrevemos a expressão no manuscrito e incluímos “vírus da raiva…” (RABV) na linha 273.
8) Você está realmente descrevendo morcegos insetívoros, não 'insetívoros' propriamente ditos, que se referiram anteriormente a um grupo informal diferente de mamíferos, como os musaranhos (Ln 284). Resposta 8 : Pedimos desculpas pela incompatibilidade. Concordamos com o comentário do revisor e reescrevemos a expressão, alterando-a para “morcegos insetívoros…” (linha 278).
9) Mudança para o uso plural de 'estes dados fornecidos' (Ln 329). Resposta 9 : Concordamos com o comentário do revisor e reescrevemos a expressão e a alteração no manuscrito (linha 361-362).
10) Se você inserir uma abreviação como RABV, use-a consistentemente em todo o MS (Ln 331). Resposta 10 : Aceitamos o comentário do revisor e atualizamos o vírus da raiva para RABV no manuscrito (linhas 71, 91, 103, 135-136, 167, 178-179, 267, 274, 313, 321, 330, 352, 363 e 370). Obrigado por apontar isso.
11) É provável que BZ consiga aplicar o NGS em estudos futuros? Resposta 11 : Acreditamos que este seja o caminho mais provável; no entanto, devido ao alto custo, esta ferramenta ficará inicialmente restrita, pelo menos num futuro próximo, a laboratórios de referência, institutos de pesquisa e universidades. Obrigado por apontar isso.
12) Já que o título se refere à "evolução" – como a raiva em saguis "evoluiu"? |
||
|
Resposta 12 : Neste artigo, realizamos uma análise abrangente apenas da raiva associada aos morcegos. Portanto, não podemos discutir a evolução do vírus da raiva mantida por tranquilidade de saguis com o conjunto de dados atuais que possuímos, uma vez que este não era o objetivo do estudo.
|
||
|
4. Resposta aos comentários sobre a qualidade da língua inglesa |
||
|
Ponto 1: O revisor apontou alguns erros de digitação e ortografia. |
||
|
Resposta: Obrigado por apontar isso. Fizemos uma nova revisão completa para corrigir esses erros.
|
||
|
5. Esclarecimentos adicionais |
||
|
Esperamos que essas melhorias sejam suficientes para concluir o processo de publicação. Permanecemos à disposição para quaisquer esclarecimentos. |
||
